# Genomic Analysis of the Suspicious SARS-CoV-2 Sequences in the Public Sequencing Database

Xiao Sun,[a,b] Chuanwen Kan,[a,b] Wentai Ma,[a,b] Zhenglin Du,[c] Mingkun Li[a,b,c]

[a]Key Laboratory of Genomic and Precision Medicine, Beijing Institute of Genomics, Chinese Academy of Sciences and China National Center for Bioinformation, Beijing, China
[b]University of Chinese Academy of Sciences, Beijing, China
[c]National Genomics Data Center, Beijing Institute of Genomics, Chinese Academy of Sciences and China National Center for Bioinformation, Beijing, China

Xiao Sun and Chuanwen Kan contributed equally to this work. Author order was determined by drawing straws.

**ABSTRACT**  SARS-CoV-2 has infected more than 600 million people. However, the origin of the virus is still unclear; knowing where the virus came from could help us prevent future zoonotic epidemics. Sequencing data, particularly metagenomic data, can profile the genomes of all species in the sample, including those not recognized at the time, thus allowing for the identification of the progenitor of SARS-CoV-2 in samples collected before the pandemic. We analyzed the data from 5,196 SARS-CoV-2-positive sequencing runs in the NCBI's SRA database with collection dates prior to 2020 or unknown. We found that the mutation patterns obtained from these suspicious SARS-CoV-2 reads did not match the genome characteristics of an unknown progenitor of the virus, suggesting that they may derive from circulating SARS-CoV-2 variants or other coronaviruses. Despite a negative result for tracking the progenitor of SARS-CoV-2, the methods developed in the study could assist in pinpointing the origin of various pathogens in the future.

**IMPORTANCE**  Sequences that are homologous to the SARS-CoV-2 genome were found in numerous sequencing runs that were not associated with the SARS-CoV-2 studies in the public database. It is unclear whether they are derived from the possible progenitor of SARS-CoV-2 or contamination of more recent SARS-CoV-2 variants circulated in the population due to the lack of information on the collection, library preparation, and sequencing processes. We have developed a computational framework to infer the evolutionary relationship between sequences based on the comparison of mutations, which enabled us to rule out the possibility that these suspicious sequences originate from unknown progenitors of SARS-CoV-2.

**KEYWORDS**  SARS-CoV-2, evolutionary relationship inference, metagenomics, progenitor, sequencing data

Since the identification of SARS-CoV-2 in late 2019, more than 633 million confirmed cases and over six million deaths have been recorded worldwide (as of 10 November 2022), posing a huge threat to global public health (1). However, the origin of the virus is still unknown, although there have been substantial discussions and a number of scientific reports have been published on the origin of SARS-CoV-2 (2, 3). Recently, a WHO-convened global study of the origins of SARS-CoV-2 proposed that the virus was highly likely to be transmitted from animals to humans through intermediate hosts or directly to humans (4). Although several close relatives of SARS-CoV-2 (RaTG13, BANAL-20-52) with nucleotide identity higher than 96% were identified in bats from Yunnan, China, and the Indochinese peninsula, Laos, the limited similarity indicates that they are unlikely to be the direct progenitor of SARS-CoV-2 infecting humans (5, 6). Previous studies mainly focused on the investigation of the SARS-like coronavirus carried by animals and vectors,

Address correspondence to Mingkun Li, limk@big.ac.cn, or Zhenglin Du, duzhl@big.ac.cn.
The authors declare no conflict of interest.

as well as environmental and clinical samples that tested positive for SARS-CoV-2 nucleic acids, which has progressed slowly due to the paucity of relevant early samples.

Metagenomic sequencing is a culture-independent method to profile all genetic material in a microbial community, which is capable of detecting both known and unknown microorganisms. The method has been applied in pathogen detection, genomic surveillance of emerging viruses, and environmental monitoring (7, 8). The large amount of metagenomic data generated by different projects provides a new dimension to look into the trajectory of the virus before the outbreak in Wuhan or human infection. A previous study identified signals of SARS-CoV-2 in the metagenomic data of fecal samples collected before April 2019 (9). However, considering that the sequencing date (unavailable in the database) is always later than the sample collection date (available in the database), it is unclear whether these SARS-CoV-2 sequences are endogenous or derived from contamination during sample processing or sequencing. Therefore, SARS-CoV-2 recovered from samples collected before the outbreak of the pandemic in late 2019 is not necessarily the progenitor of SARS-CoV-2. Thus, a computational method for verifying the legitimacy of the progenitor is highly desired. Our study analyzed all sequencing data with suspicious SARS-CoV-2 reads deposited in the NCBI's SRA (Sequence Read Archive) database. By developing a sequence-based algorithm to resolve the evolutionary relationship between sequences, we found that all viral sequences in the public database were unlikely to be the progenitor of SARS-CoV-2.

## RESULTS

**Summary of the data.** NCBI's SRA is a database collecting raw sequencing reads from all branches of life sciences. SARS-CoV-2-linked reads were detected in 350,710 sequencing runs in the database using the SRA Taxonomy Analysis Tool (as of 25 June 2021) (10). After removing the sequencing runs related to SARS-CoV-2 (with the term SARS-CoV-2/COVID-19/2019nCoV contained in the title of the BioSample/BioProject) and those sampled in 2020 and beyond, we obtained 5,196 sequencing runs which were suspected to include the SARS-CoV-2 progenitor (see Table S1 and Fig. S1 in the supplemental material), all the sequencing data were generated on the Illumina platform. The collection date of most samples (80%) was unknown, while the remaining samples were collected from 1986 to 2019 (Fig. 1A). Over half of the data (59%) was derived from samples collected in the United States, and mostly from the PulseNet program, which aims to identify the infectious agents of outbreaks, while 30% of the data had no geographical information (Fig. 1B). The host origin of most samples was unknown (61%), followed by humans (21%), animals, and the environment (Fig. 1C).

**Genome analysis of the SARS-CoV-2 reads.** There were 518 (10%) samples that had more than 1,000 reads aligned to the SARS-CoV-2 reference genome (Wuhan-Hu-1, accession number NC_045512.2) (11), which offered a 5-fold average depth (Fig. 2A). We found that these reads were evenly distributed on the viral genome, indicating that they were not likely derived from a few specific genomic regions that were homologous to SARS-CoV-2 (Fig. S2). Notably, after carefully reviewing the abstracts of relevant studies, 232 samples were suspected to be derived from SARS-CoV-2 studies and were thus discarded in the subsequent analyses.

The genome coverage (≥5-fold) reached a plateau when the read number was higher than 10,000 (Fig. 2B). Two clusters of outlier samples caught our attention (Table S2). The first data set included 20 samples with a genome coverage proportion of less than 5% despite a relatively higher number of SARS-CoV-2 reads (>1,000), which might be caused by contamination of PCR products. However, the reads were not enriched in specific regions, except for SRR13963693 and SRR13077477, indicating that they were unlikely to be derived from PCR contamination (Figure. S3). The second data set contained 23 samples with more than 100,000 SARS-CoV-2 reads but genome coverage lower than 95%, which may be due to greater divergence between these sequences and the reference sequence.

The mutation number observed in the assembled genome showed a dichotomous pattern of most samples (90%) having fewer than 20 mutations, while others (28 samples) had more than 165 mutations (Fig. 2C). We noted that all sequences in the above-

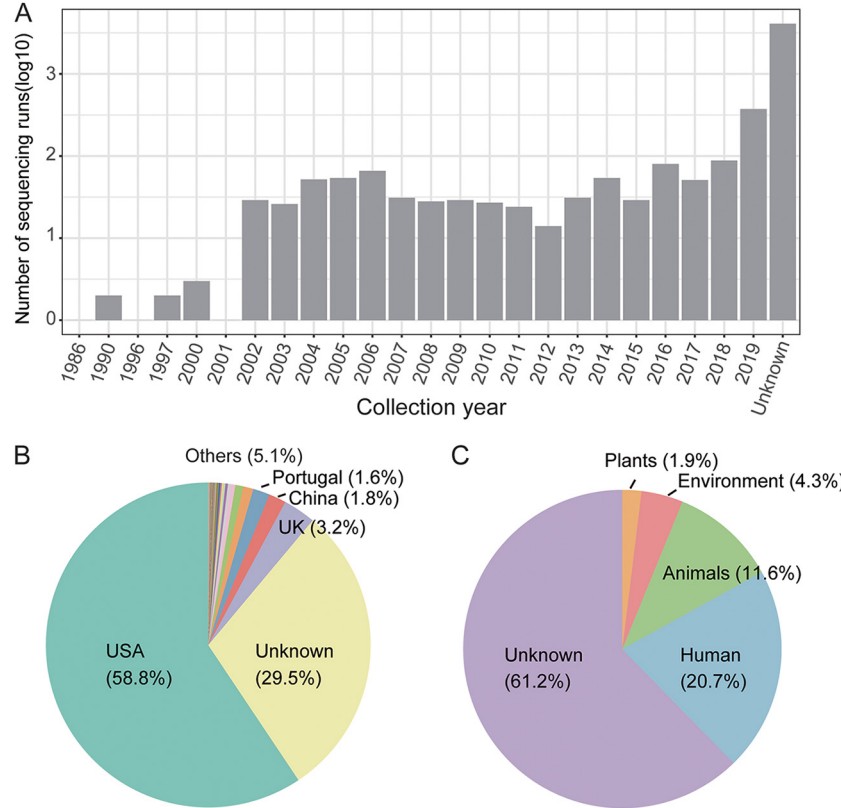

**FIG 1** Summary of 5,196 sequencing runs with suspicious SARS-CoV-2 reads. (A) The collection dates of the sequencing runs registered in the NCBI database. The sequencing runs without a collection date were classified in the unknown category. (B) The distribution of origin countries of the samples included in the sequencing runs. (C) The distribution of host sources of the samples included in the sequencing runs. The names of the five categories with the highest numbers of entries are shown in the pie chart.

mentioned second data set (poor coverage with a high number of SARS-CoV-2 reads) had more than 165 mutations, thus supporting our previous hypothesis that they were derived from a distant relative of SARS-CoV-2. Through conducting *de novo* assembly and taxonomic assignment, we found that the assembled contigs ($\geq$500 bp) came from either SARS or SARS-CoV-2 (Table S3). We noted that all contigs recognized as SARS-CoV origin were derived from studies investigating SARS-CoV, and all contigs recognized as SARS-CoV-2 were from the pangolin and bat, which have already been adequately investigated in previous studies (12–14). Therefore, the sequences with an excess number of mutations were unlikely to be the unknown progenitor of SARS-CoV-2.

**Inference of the evolutionary relationship between the suspicious SARS-CoV-2 sequences and public SARS-CoV-2 sequences.** The next challenge is to distinguish whether 258 sequences with fewer mutations ($\leq$19 mutations) were derived from the progenitor of SARS-CoV-2 or the previous circulating variants in the population. We developed a computational method to characterize the evolutionary relationship between the questioned sequence (Q) recovered in this study, the known earliest SARS-CoV-2 sequence (E) (NC_045512.2), and other available SARS-CoV-2 sequences (D) in the public database (Fig. 3A; details are described in Materials and Methods). Among 258 Q sequences, 54 sequences were identical to the E sequence (NC_045512.2); thus, their evolutionary relationship to E cannot be determined. Among the remaining 204 Q sequences, the difference between Q and E ($N_q + N_s$) was greater than the difference between Q and D ($N_d + N_q$) in 201 sequences, indicating D and Q were both descendants of E, and there were four possibilities for the relationship between D and Q: Q was identical to D ($N_q = 0$, $N_d = 0$, 135 Q), Q may be the ancestor sequence of D ($N_q = 0$, $N_d > 0$, 6 Q), D

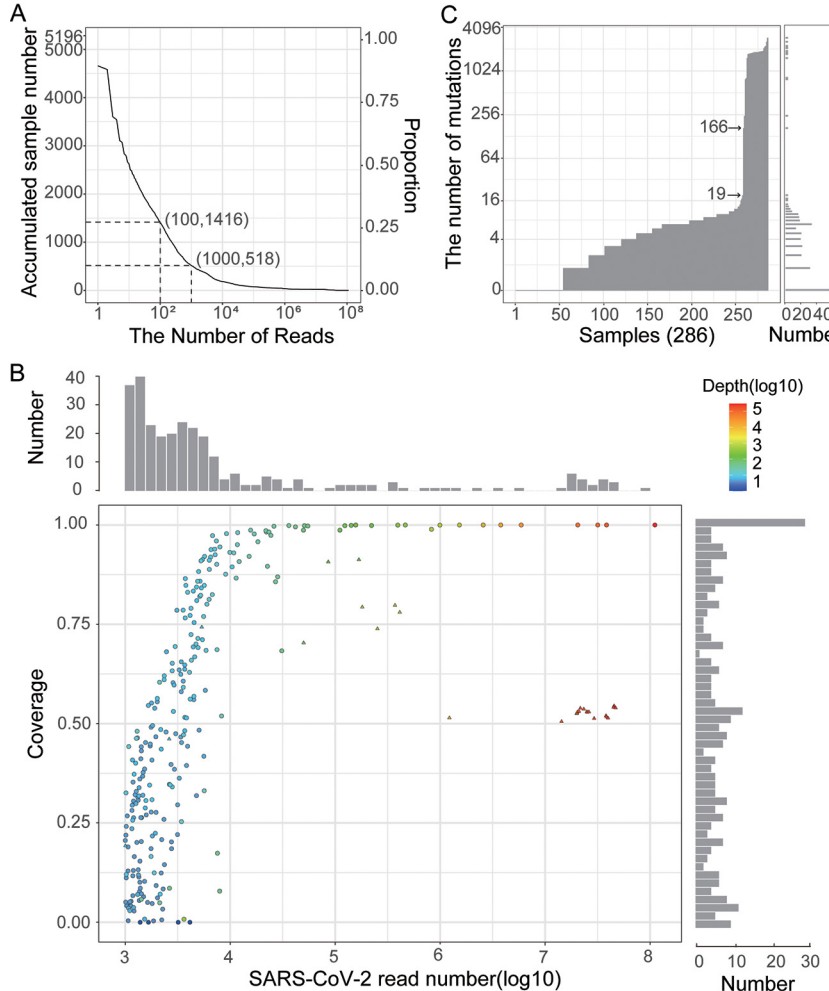

**FIG 2** Genomic characteristics of the suspicious SARS-CoV-2 sequences. (A) The number of SARS-Cov-2 reads identified in each data set. The numbers of samples with more than 100 reads and 1,000 reads are shown in the figure. (B) Genome coverage of samples with more than 1,000 SARS-CoV-2 reads. Only positions covered by at least five reads were included in the analysis. The colors represent the average sequencing depth. The two bar charts show the number of samples with particular read counts and coverage. The 28 samples with a high mutation number (>165) are depicted by triangles. (C) The number of mutations identified in each sample. The bar chart on the right shows the number of samples with particular read counts.

may be the ancestor sequence of Q ($N_q > 0$, $N_d = 0$, 56 Q), or Q and D were on evolutionarily distinct branches ($N_q > 0$, $N_d > 0$, 4 Q).

The collection date of the three suspected progenitor sequences (SRR12478493; SRR12478496; SRR13338973) was unknown (details are shown in Table S4), and each of them possessed one additional mutation relative to E (Fig. 3B), which has not been observed in public SARS-CoV-2 databases (Table 1). Then, we aligned the three sequences to distant ancestors of SARS-CoV-2 (RaTG13, GXP5L, MP789, and BANAL-20-52) (5, 6, 14, 15) and found that the mutant alleles were different from the putative ancestral alleles (Table 1), suggesting that they might not be the ancestral sequence of SARS-CoV-2. Then, we replaced NC_045512.2 with another two putative earliest SARS-CoV-2 sequences (ProCoV2, Guangdong/HKU-SZ-002/2020) (16, 17) and found no suspected progenitor sequence (Fig. S4).

Additionally, the same analysis was performed on 898 samples with SARS-CoV-2 read numbers between 100 and 1,000, and another suspected progenitor sequence was discovered in those samples ($N_q = 1$, $N_d = 1$, $N_s = 0$, SRR14613206; Table S4, Fig. S5). Again, the mutant allele possessed by the sequence was different from the nucleotide in the distant

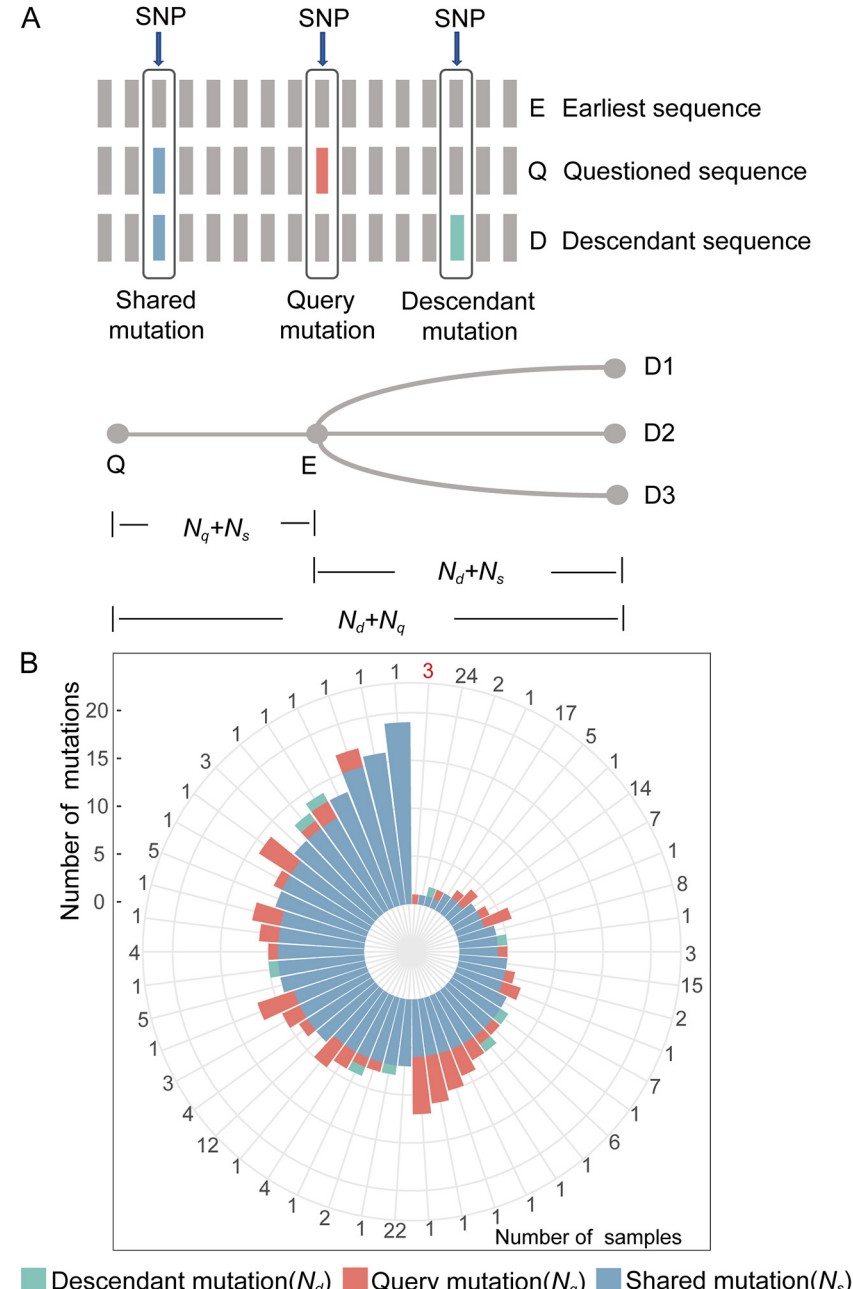

**FIG 3** Ancestral analysis of the suspicious SARS-CoV-2 sequences. (A) Schematic diagram of different types of mutations and the evolutionary tree of different sequences when Q is the progenitor of SARS-CoV-2. $N_q$ represents the number of query mutations, $N_s$ represents the number of shared mutations, and $N_d$ represents the number of descendant mutations. (B) Number of different mutations for 204 questioned sequences. NC_045512.2 was used as the earliest SARS-CoV-2 sequence (E). The outer numbers indicate the numbers of samples.

ancestral sequences (Table 1), thus not supporting the sequence as a possible progenitor. In summary, we did not identify any progenitor sequences with high confidence.

To verify the accuracy of the computational method, a simulated virus genome data set (including 248,824 sequences spanning 50 generations) with known evolutionary relationships was generated (see details in Materials and Methods). We found that the false-negative rate of the method was 0% (0 in 1,000 simulations), while the false-positive rate was 2.8% (28 in 1,000 simulations). The results are consistent with our assumption that all progenitor sequences should conform to equations 1 and 2, while not all sequences that conform to equation 1 and 2 are the progenitor sequences.

**TABLE 1** Summary of the mutation in the suspected progenitor of SARS-CoV-2

| | | | | Corresponding alleles[a] | | | | |
|---|---|---|---|---|---|---|---|---|
| Accession no. | Coverage (≥5×) | Mutation position | Mutant allele | Wuhan-Hu-1 | RaTG13 | GXP5L | MP789 | BANAL-20-52 |
| SRR12478493.1 | 0.8642 | 10132 | G | A | A | A | T | A |
| SRR12478496.1 | 0.8274 | 10132 | G | A | A | A | T | A |
| SRR13338973.1 | 0.0796 | 17294 | T | A | A | A | A | A |
| SRR14613206 | 0.0824 | 4653 | G | C | C | C | C | C |

[a]The corresponding alleles in the earliest SARS-CoV-2 sequence and distant ancestor sequences.

Moreover, the suspicious SARS-CoV-2 data have also been analyzed using the phylogenetic tree method. Due to the limited genome coverage obtained from the sequencing data (Table S2), we cannot find any overlapped genome region recovered in all samples to construct a phylogenetic tree. Instead, we filled the genome gaps by the reference sequence (NC_045512.2) as a compromise strategy. The phylogenetic tree constructed from all suspicious sequences and putative ancestral sequences (including the sequences from distant relatives, see details in Materials and Methods) indicate that one sequence (SRR14780115) might be a progenitor sequence (Fig. S6). We found that the sequence had a genome coverage of 5% before filling the gap, and there was only one mutation (C15720T) relative to NC_045512.2 in the covered region. Of note, two public SARS-CoV-2 sequences (EPI_ISL_1306112, EPI_ISL_472652) had the same sequence in the covered region, suggesting that they are unlikely to be derived from the progenitor of SARS-CoV-2.

## DISCUSSION

SARS-CoV-2 reads were detected in thousands of sequencing runs that were not associated with SARS-CoV-2 studies. It is difficult to determine if these SARS-CoV-2 reads were from the SARS-CoV-2 progenitor or contamination during sequencing, even though some of the samples were even taken before the pandemic (18). We have developed a new computational framework to solve this question by disentangling the evolutionary relationship between the sequences in question and other known SARS-CoV-2 sequences. The method assumes that the genomic difference between the progenitor sequence and the SARS-CoV-2 sequences that circulated after the outbreak is greater than that between the progenitor sequence and the known earliest SARS-CoV-2, which holds unless the backmutation took place at the early stage of the pandemic. Notably, the method aims to find as many potential progenitor sequences as possible to reduce the false-negative rate. However, it could generate some false positives when the close relative of the sequence in question is not included in the public SARS-CoV-2 database, which has been validated in our simulation data. Thus, we have further inspected all positive results by aligning the reads to the genome of distant ancestors of SRAS-CoV-2 from bats and pangolins to identify false positives. In addition, compared to the phylogenetic tree method, the new method is better suited for the incomplete genome analysis, as we discovered in this study, because the former method is normally used when there are some overlapped genomic regions between different sequences.

Our results indicated that none of the suspicious SARS-CoV-2 reads identified in the SRA database likely originated from an unidentified SARS-CoV-2 progenitor. Instead, we suspected that they might be derived from other coronaviruses or SARS-CoV-2 variants that circulated in the population after the outbreak of the pandemic, some of which were not recorded in the current public SARS-CoV-2 databases. In particular, we noted that 70 sequencing runs had suspicious SARS-CoV-2 reads submitted before 2020 (Table S5), which was unlikely to represent contamination during sampling or sequencing. Among them, 47 runs only included reads partially homologous to the SARS-CoV-2 reference genome, as no full-length read could be mapped to the SARS-CoV-2 reference genome, suggesting that no close relative of SARS-CoV-2 is included in these runs; five runs had 4 to 298 reads mapped to the SARS-CoV-2 reference genome, and no mutation was identified among a limited covered genomic region; it was unclear whether they are from a progenitor of SARS-CoV-2 or other coronaviruses. The remaining 18 samples had

more than 1 million reads mapped to the SARS-CoV-2 reference genome, and all of them are from studies of SARS and SARS-CoV-2-like viruses from bats and pangolins (Table S3). Therefore, none of the data submitted before 2020 support the presence of an unknown progenitor of SARS-CoV-2.

Failure in detecting the SARS-CoV-2 progenitor in the current sequencing data repository could be attributed to several reasons. First, the data are limited in terms of quantity and geographic representation, which might be due to the high cost and the technical barrier of metagenomic sequencing. Second, there are limited sequencing data from animals and vectors, which are more likely to carry the progenitor of SARS-CoV-2. Third, many studies only target DNA pathogens; hence RNA viruses, such as SARS-CoV-2, are not included in those data. For example, the PulseNet program, which contributed the largest number of sequencing runs in our study, aims to investigate the DNA fingerprints of the sample. Fourth, the occurrence of the backmutation at the early stage of the outbreak would violate the assumption of our method, which can result in false negatives. Therefore, a powerful metagenomic surveillance network that spans different geographic regions and hosts should be established to improve our capacity to identify emerging and reemerging pathogens, as well as the progenitor and the mode of transmission of these pathogens.

## MATERIALS AND METHODS

**Data collecting and filtering.** We retrieved all data from sequencing runs containing SARS-CoV-2 reads identified by the STAT algorithm in the NCBI's SRA database (35,0710 sequencing runs, as of 25 June 2021, https://www.ncbi.nlm.nih.gov/sra) (10). Then, the data derived from SARS-CoV-2-related studies (BioSample/BioProject title contained SARS-CoV-2, COVID-19, 2019nCoV) and those collected after 2020 were discarded.

**Genome assembly and mutation detection.** All filtered reads were aligned to NC_045512.2 using the Burrows-Wheeler MEM algorithm (BWA-MEM; v0.7.12) (11, 19). The sequencing depth, mpileup files, and read count files were retrieved from BAM files using SAMtools (v1.9; mpileup -OsBa) and VarScan (v2.3.9; –min -coverage 0) (20, 21). A mutation (including indels) was called if the frequency of the mutant allele was higher than 70% at a position covered by at least five reads. For the 28 samples with a mutation number greater than 165, de novo assembly was performed using Megahit (v1.2.9), and the taxonomy of the contigs (≥500 bp) was assigned using Kraken 2 (v2.0.8 beta; –output –report –paired) (22, 23).

**The algorithm of evolutionary relationship inference.** We classified the mutations into shared mutations (s), query mutations (q), and descendant mutations (d) to illustrate the difference between the SARS-CoV-2 sequences recovered from the sequencing runs in question (Q), the known earliest SARS-CoV-2 sequence (E), and other SARS-CoV-2 sequences in public databases (D) (Fig. 3A). If Q is the progenitor sequence of SARS-CoV-2, the difference between Q and D must be greater than the difference between Q and E; thus, the number of mutations (N) should conform to the following equations.

$$N_d + N_q > N_q + N_s \tag{1}$$

$$N_d + N_q > N_d + N_s \tag{2}$$

For a given Q, $N_q + N_s$ is a fixed value that represents the difference between Q and E. Considering a large number of D sequences in the database, our test was to examine whether the smallest $N_d + N_q$ (D with the least number of mutations from the Q sequence, which was retrieved using https://ngdc.cncb.ac.cn/ncov/online/tool/genome-tracing) satisfies equation 1. If yes, we further checked whether equation 2 was satisfied. In case any of the equations were not satisfied, we could reject the null hypothesis that Q is the progenitor of SARS-CoV-2. However, satisfying these two equations does not necessarily imply that the sequence is from the progenitor.

There are three different putative earliest SARS-CoV-2 sequences: NC_045512.2, the ProCoV2 sequence, and the Guangdong/HKU-SZ-002/2020 sequence (16, 17), which were used as E in our analysis, respectively.

**Validation of the methods using simulated sequences.** To validate the mathematical model proposed in our study, a simulation test was performed. We simulated the data for 50 generations of virus transmission (the virus genome is composed of 1,260 nucleotides that were taken from the SARS-CoV-2 N gene); each virus generated on average one offspring (following Poisson distribution) with one additional mutation generated in each offspring. The mutation rate at different positions followed the distribution as obtained in SARS-CoV-2 data. The final data set included 248,824 viral genomes.

To calculate the false-negative rate of the method, an E (known earliest viral genome) was randomly selected from the simulated data. After E was determined, Q (questioned sequence) was randomly selected from the direct ancestors or the collateral ancestors (siblings of the direct ancestors) of E. Given that in the real world it would be impossible to obtain all descendant sequences of E, we selected 10% of the descendant sequences randomly as the D (public viral genomes) sequence database. The process was replicated 1,000 times, and the false-negative rate was calculated as the chance that Q did not satisfy equations 1 and 2. In contrast, when calculating the false-positive rate of the method, Q was randomly selected from the direct

descendants or the collateral descendants (siblings of the direct descendants) of E. Similarly, 10% of the descendant sequences were randomly selected as the D sequence database. The process was replicated 1,000 times, and the false-positive rate of the method was calculated as the chance that Q satisfied equations 1 and 2.

**Construction of the phylogenetic tree.** First, the uncovered genomic regions of the suspicious SARS-CoV-2 sequences were filled with the SARS-CoV-2 reference genome (NC_045512.2). Then we constructed a phylogenetic tree using all filled suspicious sequences and putative ancestral sequences of SARS-CoV-2 (RaTG13, GXP5L, MP789, and BANAL-20-52). Multiple-sequence alignment was performed using MAFFT (v7.453; –auto), and the phylogenetic tree was constructed with the maximum likelihood method implemented in IQ-TREE (v2.2.1; -m GTR+F -asr) (24, 25).

**Data availability.** Data and scripts to replicate all analyses are available at https://github.com/program-ancestral-determination/Ancestral-determination.

## SUPPLEMENTAL MATERIAL

Supplemental material is available online only.
**SUPPLEMENTAL FILE 1**, PDF file, 1.3 MB.
**SUPPLEMENTAL FILE 2**, XLSX file, 0.1 MB.

## ACKNOWLEDGMENTS

We thank Yinying Wang and graduate students from Beijing Institute of Genomics, Chinese Academy of Sciences for their assistance in data curation.

This study was funded by the National Natural Science Foundation of China (grant no. 82161148009), the Strategic Priority Research Program of Chinese Academy of Sciences, China (grant no. XDB38030400), and the Capital Health Development and Research Special Programme (grant no. 2021-1G-3012).

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
