## [Reviewer comments · Microbiology Spectrum]

Microbiology Spectrum

Genomic analysis of the suspicious SARS-CoV-2 sequences in the public sequencing database

Xiao Sun, Chuanwen Kan, Wentai Ma, Zhenglin Du, and Mingkun Li

Corresponding Author(s): Mingkun Li, Beijing institute of genomics, Chinese Academy of Sciences

Review Timeline:

Submission Date:	August 29, 2022
Editorial Decision:	September 20, 2022
Revision Received:	November 15, 2022
Accepted:	December 13, 2022

Editor: Feng Gao

Reviewer(s): Disclosure of reviewer identity is with reference to reviewer comments included in decision letter(s). The following individuals involved in review of your submission have agreed to reveal their identity: Aiping Wu (Reviewer #1)

Transaction Report:

DOI: <https://doi.org/10.1128/spectrum.03426-22>

September 20, 2022

Prof. Mingkun Li
Beijing Institute of Genomics, Chinese Academy of Sciences
NO.1 Beichen West Road, Chaoyang District
Beijing 100101
China

Re: Spectrum03426-22 (Genomic analysis of the suspicious SARS-CoV-2 sequences in the public sequencing database)

Dear Prof. Mingkun Li:

Link Not Available

Sincerely,

Feng Gao

Journals Department
Reviewer comments:

Reviewer #1 (Comments for the Author):

The authors have proposed a strategy to identify the progenitor of SARS-CoV-2 from the sequencing metagenomic samples collected before the pandemic. They analyzed the data from 5196 SARS-CoV-2 positive sequencing run in the NCBI's SRA database with the collection data prior to 2020 or unknown, and achieved the negative conclusion. Though it's still unclear for the progenitor of SARS-CoV-2, the tracking strategy postposed in this work is interesting in shading light on finding the potential origin of pathogens in the future. Some concerns should be addressed.

Major concerns:

1. In the "Ancestry inference of the SARS-CoV-2 sequence", the authors mentioned that they removed 54 sequences which

were identical to the E sequences. Why had these sequences been removed? What's the hypothesis underlying this step?

2. In the Discussion part, more detailed discussion are expected for the reasonability of method for tracking the potential origin of SARS-CoV-2 based on the SNP numbers.
3. For the algorithm of ancestry inference, the absolute of SNPs are used. I'm curious if the method could be biased by the sequencing coverage and GAPS. The second question is that whether the inferring results could be further evidenced by traditional phylogenetic analysis?

Minor concerns:

1. The Importance is highly redundant to Abstract, which should be re-organized.
2. Some statements in the manuscript should be supported with appropriate references, such as the first sentence in Background, "more than 600 million confirmed cases and six million deaths..."

Reviewer #2 (Comments for the Author):

The publication entitled "Genomic analysis of the suspicious SARS-CoV-2 sequences in the public sequencing database" by Sun and Kan et al, present a study based on metagenomics datasets trying to find sequences that correspond to SARS-CoV-2 before the appearance of the pandemic. To know if these questioned sequences are more ancestral than the first described SARS-CoV-2 sequences, they test if the mutations in the questioned sequences are more divergent between the genome ancestors than the known SARS-CoV-2 descendants. The importance of this study lies in a first objective: analyzing pre-pandemic metagenomic reads belonging to SARS-CoV-2 and second: to know if they are ancestral sequences. To achieve these two mentioned objectives, I think that some questions must be answered to clarify this study.

To clarify the first objective is to ensure that the suspicious metagenomic runs must be uploaded before the pandemic (see major comments 4 and 5), and they are really prepandemic, also in this way it can be solved the problem of laboratory contamination in the sample. And referring to the second objective I do not see any novelty using this method instead a phylogenetic reconstruction (see major comment 8), also this method proposed should be tested in all possible ways doing a test accuracy with all possibilities and comparing this method to other methodologies such as a phylogeny with all sequences (see major comments 7 and 8).

Major comments:

1. There is no caption/description of supplementary figures of this work.
2. It is difficult to follow at each moment what sequence datasets are used and why they are discarded. To be able to follow in the text I propose to identify them in some way, for example to make a supplementary figure with the applied filters and the sets of runs/sequences derived from them. (L97-98, L101-103, L105-108, L110-111)
3. What collection date are the selected datasets from? (L97-98, L101-103, L105-108, L110-111)
4. The selection methods to obtain the 5,196 sequencing runs in lines 85-88 mention that metagenomic runs that are later than 2020 (included) are discarded, but in Figure 1 it is said that the vast majority are of unknown date. To be sure that they do not belong to the pandemic of SARS-CoV-2, those whose date is unknown must be discarded.
5. In table S1, I believe that all the known dates of both, the collection date sample and the upload should be given. Although the exact date on which the sample was sequenced is not known, the date of submission before 2020 will ensure they are prepandemic.
6. In line 128, to be a computational method I hope that a program will be shown in which the mathematical method presented can be replicated and tested. There is no program developed by the authors.
7. I believe that for the mathematical model to be valid a significant number of replications should be carried out where all possible results are tested. In addition, test whether there are false positives or false negatives. Also, it is important to add to this model corresponding statistical tests to reject the null hypothesis.
8. This method needs to be compared with other methods to infer the relationships between the sequences and know which ones are ancestrally enclaved. I do not understand what the improvement is of using this method compared to carrying out a phylogeny, where different methods and inferences are used taking into account different models of evolution.
9. In the keywords ancestral inference is mentioned but I do not see that inferences of the ancestral state are made in this work.
10. Lines 184-185: What sequencing techniques do the reads that are analyzed come from?
11. Lines 198-200: For a position to be considered a mutation, what is it compared against? Against what reference? Are indels considered?

Minor comments:

1. I think that the description of figure 1 should be improved, since it is a very simple description. In the pie plot the percentages should be added next to the tags or add the number to which it corresponds.
2. check line 306 "reads is marked"
3. Line 49: From when and until when is this data, a range of millions of deaths should be given since it is an estimate. Where is the reference for the estimation of deaths?
4. Lines 67-69, "Previous studies" only one reference is given.
5. In line 126, which are the sequences referred to by "these sequences" ?

Staff Comments:

Preparing Revision Guidelines

Please return the manuscript within 60 days; if you cannot complete the modification within this time period, please contact me. If you do not wish to modify the manuscript and prefer to submit it to another journal, please notify me of your decision immediately so that the manuscript may be formally withdrawn from consideration by Microbiology Spectrum.

The publication entitled "Genomic analysis of the suspicious SARS-CoV-2 sequences in the public sequencing database" by Sun and Kan et al, present a study based on metagenomics datasets trying to find sequences that correspond to SARS-CoV-2 before the appearance of the pandemic. To know if these questioned sequences are more ancestral than the first described SARS-CoV-2 sequences, they test if the mutations in the questioned sequences are more divergent between the genome ancestors than the known SARS-CoV-2 descendants. The importance of this study lies in a first objective: analyzing pre-pandemic metagenomic reads belonging to SARS-CoV-2 and second: to know if they are ancestral sequences. To achieve these two mentioned objectives, I think that some questions must be answered to clarify this study.

To clarify the first objective is to ensure that the suspicious metagenomic runs must be uploaded before the pandemic (see major comments 4 and 5), and they are really pre-pandemic, also in this way it can be solved the problem of laboratory contamination in the sample. And referring to the second objective I do not see any novelty using this method instead a phylogenetic reconstruction (see major comment 8), also this method proposed should be tested in all possible ways doing a test accuracy with all possibilities and comparing this method to other methodologies such as a phylogeny with all sequences (see major comments 7 and 8).

Major comments:

1. There is no caption/description of supplementary figures of this work.
2. It is difficult to follow at each moment what sequence datasets are used and why they are discarded. To be able to follow in the text I propose to identify them in some way, for example to make a supplementary figure with the applied filters and the sets of runs/sequences derived from them. (L97-98, L101-103, L105-108, L110-111)
3. What collection date are the selected datasets from? (L97-98, L101-103, L105-108, L110-111)
4. The selection methods to obtain the 5,196 sequencing runs in lines 85-88 mention that metagenomic runs that are later than 2020 (included) are discarded, but in Figure 1 it is said that the vast majority are of unknown date. To be sure that they do not belong to the pandemic of SARS-CoV-2, those whose date is unknown must be discarded.
5. In table S1, I believe that all the known dates of both, the collection date sample and the upload should be given. Although the exact date on which the sample was sequenced is not known, the date of submission before 2020 will ensure they are pre-pandemic.
6. In line 128, to be a computational method I hope that a program will be shown in which the mathematical method presented can be replicated and tested. There is no program developed by the authors.
7. I believe that for the mathematical model to be valid a significant number of replications should be carried out where all possible results are tested. In addition, test whether there are false positives or false negatives. Also, it is important to add to this model corresponding statistical tests to reject the null hypothesis.
8. This method needs to be compared with other methods to infer the relationships between the sequences and know which ones are ancestrally enclaved. I do not

understand what the improvement is of using this method compared to carrying out a phylogeny, where different methods and inferences are used taking into account different models of evolution.

9. In the keywords ancestral inference is mentioned but I do not see that inferences of the ancestral state are made in this work.

10. Lines 184-185: What sequencing techniques do the reads that are analyzed come from?

11. Lines 198-200: For a position to be considered a mutation, what is it compared against? Against what reference? Are indels considered?

Minor comments:

1. I think that the description of figure 1 should be improved, since it is a very simple description. In the pie plot the percentages should be added next to the tags or add the number to which it corresponds.

2. check line 306 "reads is marked"

3. Line 49: From when and until when is this data, a range of millions of deaths should be given since it is an estimate. Where is the reference for the estimation of deaths?

4. Lines 67-69, "Previous studies" only one reference is given.

5. In line 126, which are the sequences referred to by "these sequences" ?

Dear editor,

We thank the reviewers for the insightful comments and suggestions. We have modified the manuscript accordingly. We hope that these changes have resulted in an improved manuscript. Please find below our point-by-point responses to the issues raised by the reviewers. Our responses are highlighted in blue. Data and scripts used to replicate the analyses are available on the Github website (<https://github.com/program-ancestral-determination/Ancestral-determination>). Please note that the line number are marked according to the Marked Up Manuscript

Responses to reviewer #1:

Major concerns:

1. In the "Ancestry inference of the SARS-CoV-2 sequence", the authors mentioned that they removed 54 sequences which were identical to the E sequences. Why had these sequences been removed? What's the hypothesis underlying this step?

Response: Thanks for the comments. If the questioned sequence (that recovered from the SRA database) is identical to the known earliest SARS-CoV-2 sequence (E), the algorithm has no power to determine its evolutionary relationship to E. Thus, our subsequent analyses only focused on questioned sequences that are different from E. Notably, three putative earliest SARS-CoV-2 sequences (E) were used in the analysis (Wuhan-Hu-1, ProCoV2, and Guangdong/HKU-SZ-001/2020), all sequences were included in the analysis when new E was applied, for example, those 54 sequences

were included in the analysis when ProCoV2 was used as E. To avoid misunderstanding, we rephrased the sentence to “54 sequences were identical to the E sequence (NC_045512.2), thus their evolutionary relationship to E cannot be determined.” (line 133).

2. In the Discussion part, more detailed discussion are expected for the reasonability of method for tracking the potential origin of SARS-CoV-2 based on the SNP numbers.

Response: Thanks for the comments. The principle and the limitation of the method have been discussed in the revised manuscript (lines 181-191). Moreover, simulation validation and comparison with the phylogenetic method have been added in the revised manuscript (lines 156-175).

3. For the algorithm of ancestry inference, the absolute of SNPs are used. I'm curious if the method could be biased by the sequencing coverage and GAPS. The second question is that whether the inferring results could be further evidenced by traditional phylogenetic analysis?

Response: Thanks for the comments. We agree that the performance of the method would be influenced by the sequencing coverage and gaps, as higher coverage and fewer gaps would result in higher accuracy. Thus, our analysis focused on samples with a relatively high sequencing depth (with more than 1,000 reads aligned to the SARS-CoV-2). However, there are still many incomplete genomes recovered from the reads, i.e., 81% of samples have less than 90% genome covered, and 53% of samples have less than 50% genome covered. A new supplementary table 2 that provides the

genome coverage of all samples has been added to the revised manuscript.

Due to the limited genome coverage obtained from the sequencing data, it is tricky to apply the traditional phylogenetic analysis because we cannot find any overlapped genome region recovered in all samples to construct a phylogenetic tree. A compromised solution is to fill the gaps by the reference sequence. The phylogenetic tree constructed from all suspicious sequences and putative ancestral sequences (including the sequences from distant relatives) indicated that one sequence (SRR14780115) might be a progenitor sequence (Figure S6). The sequence has a genome coverage of 4% before filling the gap, and has only one mutation (C15720T) relative to the Wuhan-Hu-1 (NC_045512.2). Notably, there are two public SARS-CoV-2 sequences (EPI_ISL_1306112, EPI_ISL_472652) that have the same sequence in the covered region, suggesting that it is unlikely to be an ancestral sequence. The results of the phylogenetic tree method have now been added in the revised version of the manuscript (lines 163-175), and the comparison of our method with the phylogenetic tree method is discussed in the revised manuscript (lines 193-196).

Minor concerns:

1. The Importance is highly redundant to Abstract, which should be re-organized.

Response: Thanks for your comments. We have rewritten the Importance in the revised manuscript. Please find below the new paragraph.

Importance: Sequences that are homologous to the SARS-CoV-2 genome were found in numerous sequencing runs that were not associated with the SARS-CoV-2 studies

in the public database. It is unclear whether they are derived from the possible progenitor of SARS-CoV-2 or contamination of more recent SARS-CoV-2 variants circulated in the population due to the lack of information on the collection, library preparation, and sequencing processes. Here, we have developed a computational framework to infer the evolutionary relationship between sequences based on the comparison of mutations, which enabled us to rule out the possibility that these suspicious sequences originate from unknown progenitors of SARS-CoV-2.

2. Some statements in the manuscript should be supported with appropriate references, such as the first sentence in Background, "more than 600 million confirmed cases and six million deaths...".

Response: Thanks for the suggestion, we have added the references to the statement in the revised manuscript.

Responses to reviewer #2:

Major comments:

1. There is no caption/description of supplementary figures of this work.

Response: Thanks to the reviewer for the reminder. We have previously included the caption and description of the supplementary figures in a supplementary Word file. Now we have included this information together with the supplementary figures in the supplemental file1 according to the requirement of the journal.

2. It is difficult to follow at each moment what sequence datasets are used and why they are discarded. To be able to follow in the text I propose to identify them in

some way, for example to make a supplementary figure with the applied filters and the sets of runs/sequences derived from them. (L97-98, L101-103, L105-108, L110-111)

Response: Thanks for the suggestion. A flowchart has been added as Figure S1 to show how the data was processed in the revised manuscript (also attached below).

Supplementary Figure 1. The flowchart of data filtering and analysis.

3. What collection date are the selected datasets from? (L97-98, L101-103,

L105-108, L110-111)

Response: We have added the collection date of all sequences in the revised Supplementary Table 1. In addition, the answers to the question are shown below.

(1) The collection date of 518 samples that had more than 1,000 reads aligned to SARS-CoV-2 reference genome (NC_045512.2) (mentioned in L97-98 in the old version) is shown in the following table.

Collection date	Number of samples
2005	3
2006	4
2014	2
2016	8
2017	16
2018	3
2019	14
Unknown	468

(2) The collection date of 232 samples which are suspected to be derived from SARS-CoV-2 studies (mentioned in L101-103 in the old version) is unknown.

(3) The collection date of 20 samples with genome coverage proportion less than 5% (mentioned in L105-108 in the old version) is shown in the following table.

Collection date	Number of samples
2016	4
2018	1

2019	2
Unknown	13

(4) The collection date of 23 samples with more than 100,000 SARS-CoV-2 reads but genome coverage lower than 95% (mentioned in L110-111 in the old version) is shown in the following table.

Collection date	Number of samples
2017	4
2019	1
Unknown	18

The accession numbers of the above datasets are labelled in Supplementary Table 1.

4. The selection methods to obtain the 5,196 sequencing runs in lines 85-88 mention that metagenomic runs that are later than 2020 (included) are discarded, but in Figure 1 it is said that the vast majority are of unknown date. To be sure that they do not belong to the pandemic of SARS-CoV-2, those whose date is unknown must be discarded.

Response: Thanks for the comments from the reviewer. We agree that more stringent criteria will reduce the false positive rate. First, sequencing runs associated with SARS-CoV-2 studies and those with collection dates in 2020 and later have been discarded in the analysis. Meanwhile, although we also planed to remove the data generated in 2020 and later to avoid the false positive caused by the contamination during sampling or sequencing, the sequencing date is not provided in the database, and it could be significantly different from the collection date and submission date.

Second, for those sequence runs with unknown collection dates and not associated with SARS-CoV-2 studies, it is unclear whether they were collected during the pandemic or before the pandemic. A similar problem exists when using the submission date, which is discussed in the response to the next question. To avoid false negatives, we decided to include these sequences in our study. For the putative progenitor candidate recovered from sequencing runs with an unknown collection date, we have provided the collection date and submission date in the revised manuscript (Supplementary Table 4). And a new paragraph discussing the sequencing runs submitted before 2020 has been added to the revised manuscript (lines 201-213). Besides, as no potential progenitor was identified in our study, excluding the sequencing runs with unknown collection dates would not influence our conclusion.

5. In table S1, I believe that all the known dates of both, the collection date sample and the upload should be given. Although the exact date on which the sample was sequenced is not known, the date of submission before 2020 will ensure they are pre-pandemic.

Response: Thanks for the suggestion of the reviewer. We have included the collection date and submission date of the sequencing runs in the revised Supplementary Table 1. In addition, we agree that an alternative filtering strategy is to make use of the submission date which is available for all sequencing runs instead of the collection date which is unavailable for many sequencing runs. However, we found that the submission date could be significantly different from the collection date, e.g., some sequencing data were submitted 30 years after the collection date. Many sequencing

runs with collection dates before 2020 were submitted in 2020 and later. Thus, we decided to include the data submitted in 2020 and beyond to avoid false negatives. For the putative progenitor of the SARS-CoV-2, their collection and submission date have been added in the revised manuscript (Supplementary Table 4).

Meanwhile, if we remove all sequencing runs submitted after 2019 (which is the most stringent criterion) following the suggestion of the reviewer, only 70 sequencing runs had suspicious SARS-CoV-2 reads (Table S5). Among them, 47 samples had no read mapped to the SARS-CoV-2 reference genome (in the full length), suggesting a low similarity between the suspicious SARS-CoV-2 reads and the SARS-CoV-2 reference genome; Five samples had 4 to 298 reads mapped to the SARS-CoV-2 reference genome, and no mutation was identified in limited covered genomic regions; The remaining 18 samples had more than one million reads mapped to the SARS-CoV-2 reference genome, and all of them are from studies of SARS and SARS-CoV-2-like viruses from bats and pangolins (Table S3). Therefore, none of the data submitted before 2020 support the presence of an unknown progenitor of SARS-CoV-2. A discussion and a new Table S5 have been added to the revised manuscript (lines 201-213).

6. In line 128, to be a computational method I hope that a program will be shown in which the mathematical method presented can be replicated and tested. There is no program developed by the authors.

Response: Following the suggestion of the reviewer, we have uploaded the files and scripts that are necessary to replicate our analysis in Github

(<https://github.com/program-ancestral-determination/Ancestral-determination>).

7. I believe that for the mathematical model to be valid a significant number of replications should be carried out where all possible results are tested. In addition, test whether there are false positives or false negatives. Also, it is important to add to this model corresponding statistical tests to reject the null hypothesis.

Response: Thanks for the comments. To validate the mathematical model proposed in our study, we conducted a simulation test. We simulated the data for 50 generations of virus transmission (the virus genome is composed of 1260 nucleotides that were taken from the SARS-CoV-2 N gene), each virus generated on average one offspring (following Poisson distribution) with one additional mutation generated in each offspring. The mutation rate at different positions followed the distribution as obtained in SARS-CoV-2 data. The final dataset included 248,824 viral genomes.

To calculate the false negative rate of the method, an E (known earliest viral genome) was randomly selected from the simulated data. After E was determined, Q (questioned sequence) is randomly selected from the direct ancestors or the collateral ancestors (siblings of the direct ancestors) of E. Given that in the real world it would be impossible to obtain all descendant sequences of E, we selected 10% of the descendant sequences randomly as D (public viral genomes) sequence database. The false negative rate was calculated as the chance that Q did not satisfy Equations 1 and 2. In contrast, when calculating the false positive rate of the method, Q is randomly selected from the direct descendants or the collateral descendants (siblings of the direct descendants) of E. Similarly, 10% of the descendant sequences were randomly

selected as D sequence database. The false positive rate of the method was calculated as the chance that Q satisfied Equations 1 and 2. By replicating the aforementioned process 1000 times, we obtained a false negative rate of 0% and a false positive rate of 2.8%. The results complied with our hypothesis: If Q is the progenitor sequence of the virus, the number of mutations should conform to Equations 1 and 2 (no false negative). In contrast, if Q is not the progenitor sequence of SARS-CoV-2, the number of mutations may also conform to Equations 1 and 2 due to the absence of a close relative of Q in the D database (some false positives).

We have included the simulation results in the revised manuscript (lines 156-162), the simulation data and the scripts to replicate the analyses are available at <https://github.com/program-ancestral-determination/Ancestral-determination>.

Meanwhile, the principle and the limitation of the method have been discussed in the revised manuscript (lines 181-191). Unfortunately, given a specific Q, we cannot calculate a statistic to represent how reliable the inference is, as the performance of the method is dependent on the completeness of the D database, such limitation has been discussed in the revised manuscript (lines 189-191).

8. This method needs to be compared with other methods to infer the relationships between the sequences and know which ones are ancestrally enclaved. I do not understand what the improvement is of using this method compared to carrying out a phylogeny, where different methods and inferences are used taking into account different models of evolution.

Response: The principle of the method is that the genomic difference between the

progenitor sequence and the SARS-CoV-2 sequences that circulated after the outbreak should be greater than that between the progenitor sequence and the known earliest SARS-CoV-2, which in theory is accurate unless the backmutation took place at the early stage of the pandemic. Validation of the method using simulated data has been included in the revised manuscript (lines 181-191).

Concerning the comparison with the phylogeny method, considering the limited genome coverage obtained from the sequencing runs (81% of samples have less than 90% genome covered, and 53% of samples have less than 50% genome covered, Supplementary Table 2), it is tricky to apply the traditional phylogenetic analysis because we cannot find any overlapped genomic region recovered in all samples to construct a phylogenetic tree. A compromised solution is to fill the gaps by the reference sequence. Under which circumstances, the phylogenetic tree constructed from all suspicious sequences and putative ancestral sequences (including the sequences from distant relatives) indicated that one sequence (SRR14780115) might be a progenitor sequence (Figure S6). The sequence has a genome coverage of 5% before filling the gaps, and has only one mutation (C15720T) relative to the Wuhan-Hu-1(NC_045512.2). Notably, there are two public SARS-CoV-2 sequences (EPI_ISL_1306112, EPI_ISL_472652) that have the same sequence in the covered region, suggesting that it is unlikely to be an ancestral sequence. The results of the phylogenetic tree method have been added in the revised manuscript (lines 163-175), and the comparison of our method with the phylogenetic tree method is discussed in the revised manuscript (lines 193-196).

9. In the keywords ancestral inference is mentioned but I do not see that inferences of the ancestral state are made in this work.

Response: Sorry for the confusion, ancestral inference means inference of the evolutionary relationship between the suspicious SARS-CoV-2 sequences and public SARS-CoV-2 sequences. We have rephrased the word to “evolutionary relationship inference” to avoid misleading and revised the subtitle of the third section in the Result to “Inference of the evolutionary relationship between the suspicious SARS-CoV-2 sequences and public SARS-CoV-2 genomes” in the revised manuscript.

10. Lines 184-185: What sequencing techniques do the reads that are analyzed come from?

Response: All the reads analyzed in the study are generated on the Illumina platform. This information has been added in lines 88-89 in the revised manuscript

11. Lines 198-200: For a position to be considered a mutation, what is it compared against? Against what reference? Are indels considered?

Response: In the study, the mutation was called using NC_045512.2 (Wuhan-Hu-1) as the reference sequence. The mutation calling pipeline was described in Methods (lines 236-240). Indels were included in the analysis, which has been specified in line 239 in the revised manuscript.

Minor comments:

1. I think that the description of figure 1 should be improved, since it is a very simple description. In the pie plot the percentages should be added next to the tags

or add the number to which it corresponds.

Response: Thank you for the suggestion. We have rephrased the description of Figure 1 to “A. The collection date of the sequencing runs registered in the NCBI database. The sequencing runs without collection date were classified into the Unknown category. B. The distribution of origin countries of the samples included in the sequencing runs. C. The distribution of host sources of the samples included in the sequencing runs. The names of the five categories with the highest number of entries are shown in the pie chart”. Meanwhile, we have included the percentages of each category next to the tags in the pie chart following the reviewer’s suggestion.

2. check line 306 "reads is marked"

Response: We have rephrased this sentence to “The numbers of samples with more than 1000 and 100 SARS-CoV-2 reads are labelled in the figure”.

3. Line 49: From when and until when is this data, a range of millions of deaths should be given since it is an estimate. Where is the reference for the estimation of deaths?

Response: Thanks for the comments, the sentence has been rephrased to “more than 633 million confirmed cases and over six million deaths have been recorded worldwide (until November 10, 2022)”, and a reference (the reference paper of <https://coronavirus.jhu.edu/map.html>) has been added in the revised manuscript.

4. Lines 67-69, "Previous studies" only one reference is given.

Response: Thanks for your reminder. The sentence has been rephrased to “A previous study identified signals of SARS-CoV-2 in the metagenomic data of fecal samples

collected before April 2019” in the revised manuscript.

5. In line 126, which are the sequences referred to by "these sequences" ?

Response: Sorry for the confusion. These sequences refer to 258 sequences with less than 20 mutations. We have rephrased the sentence to “The next challenge is to distinguish whether 258 sequences with fewer mutations (< 20 mutations) were derived from the progenitor of SARS-CoV-2 or the previous circulating variants in the population”, and we believe that it will be much clearer as a flowchart has been added as Figure S1 to show how the data was processed in the revised manuscript.

December 13, 2022

Prof. Mingkun Li
Beijing institute of genomics, Chinese Academy of Sciences
NO.1 Beichen West Road, Chaoyang District
Beijing 100101
China

Re: Spectrum03426-22R1 (Genomic analysis of the suspicious SARS-CoV-2 sequences in the public sequencing database)

Dear Prof. Mingkun Li:

Your manuscript has been accepted, and I am forwarding it to the ASM Journals Department for publication. You will be notified when your proofs are ready to be viewed.

Sincerely,

Feng Gao
Editor, Microbiology Spectrum

Journals Department
Supplemental file1: Accept
Supplemental file2: Accept